# BHGAttN: A Feature-Enhanced Hierarchical Graph Attention Network for Sentiment Analysis

**DOI:** 10.3390/e24111691

**Published:** 2022-11-18

**Authors:** Junjun Zhang, Zhengyan Cui, Hyun Jun Park, Giseop Noh

**Affiliations:** 1Department of Computer Information Engineering, Cheongju University, Cheongju 28503, Republic of Korea; 2Division of Software Convergence, Cheongju University, Cheongju 28503, Republic of Korea

**Keywords:** sentiment analysis, BERT intermediate layer, hierarchical information encoding, hierarchical graph attention network

## Abstract

Recently, with the rise of deep learning, text classification techniques have developed rapidly. However, the existing work usually takes the entire text as the modeling object and pays less attention to the hierarchical structure within the text, ignoring the internal connection between the upper and lower sentences. To address these issues, this paper proposes a Bert-based hierarchical graph attention network model (BHGAttN) based on a large-scale pretrained model and graph attention network to model the hierarchical relationship of texts. During modeling, the semantic features are enhanced by the output of the intermediate layer of BERT, and the multilevel hierarchical graph network corresponding to each layer of BERT is constructed by using the dependencies between the whole sentence and the subsentence. This model pays attention to the layer-by-layer semantic information and the hierarchical relationship within the text. The experimental results show that the BHGAttN model exhibits significant competitive advantages compared with the current state-of-the-art baseline models.

## 1. Introduction

The sentiment analysis task is one of the most classic tasks in NLP and plays a very important role in the field of NLP research. Early sentiment analysis methods mainly include traditional machine learning algorithms such as SVM [1], k-nearest neighbor, naive Bayes [2], etc. These methods are simple to implement and have high prediction accuracy and have achieved effective results in sentiment analysis tasks. However, these methods rely heavily on domain knowledge, and the text representation is high-latitude and sparse, and the feature expression ability is weak, which shows serious shortcomings in large-scale sample training. In recent years, the rapid development of deep learning has successfully promoted the research of sentiment analysis technology. The traditional learning algorithms relying on feature engineering have been completely changed by various end-to-end deep learning. The TextCNN model proposed by Kim [3] has obvious advantages in capturing local features. TextRNN [4] and its variant models [5,6] have short-term memory and can better express contextual information. However, these models cannot model longer sequence information and are not effective in dealing with long-range dependencies.

In recent years, with the emergence of transformers [7], large-scale pretraining models with attention mechanisms such as the core GPT [8,9,10], T5 [11], BERT [12], etc., have successively refreshed many NLP fields, and more and more researchers have begun to pay attention to the application of large-scale pre-training models. However, due to the complexity of natural language structure, the above methods usually model the entire text, and consider less the semantic structure inside the text. However, in the practice of sentiment analysis, there are many mixed emotions in many texts. For example, a sentence has a positive emotion, a neutral emotion, and a negative emotion. If the semantic modeling of mixed-emotion sentences with multiple emotional tendencies is directly carried out on the whole, it may increase the difficulty of emotion judgment of the emotion analysis model, which makes it difficult for the model to be applied to the classification of mixed-emotion sentences. Intuitively, if the structural relationship of the sentence is considered, it will help the judgment of emotional polarity.

Recently, GNNs [13,14] have been shown to have strong representational capabilities in modeling structural information. The TextGCN model proposed by Yao et al. [15] builds a heterogeneous graph based on the relationship between documents and words, enabling the semisupervised classification of text on GCN. Huang et al. [16] improved the TextGCN model and proposed to generate a graph for each text, which saves memory while ensuring the effect is improved. Lin et al. [17] proposed to take advantage of large-scale pretraining models and graph networks to embed nodes in text graphs with BERT for initial word embedding, and then jointly train BERT and GCN modules to influence the representation of training data and unlabeled test data, achieving SOFT results on a wide range of text classification datasets. However, there are still some deficiencies in the current research on the construction graph method. For example, once the graph based on the global structure of TextGCN [15] is established, it cannot dynamically perceive the structural information inside a single document according to context semantics. Compared with the graph based on the global structure, TLGNN [16] can better learn the word-level relationship within a single text, but it does not pay attention to the dependencies between sentences within the document and cannot capture the structural information within the sample.

Based on the above analysis, we propose a novel modeling approach based on a large-scale pretrained model—BERT—and a hierarchical graph network. Different from the previous work, on the one hand, we use the language knowledge of the hidden state in the middle of BERT to enhance the semantic representation, and propose a BERT-based hierarchical information encoding method. This is different from the previous hierarchical coding model. For example, HAN [18] adopts the strategy of coding each level separately and then merging. This approach ignores the influence of the overall context when encoding the semantic information of subsentences. The pretrained language model BERT is able to pay attention to the correlation between the information at the subsentence level due to the task designed by NSP. At the same time, since BERT is a multilayer bidirectional encoder, the granularity of information extracted by BERT increases with the increase in the number of layers. By using each layer, more information can be fully introduced to enhance the semantic representation. On the other hand, in order to better learn the hierarchical relationship between sentences within the text, we consider that it is more reasonable to combine BERT’s layers with the hierarchy of the text for modeling. The constructed model not only focuses on the semantic information layer by layer, but also the hierarchical relationship between sentences.

To the best of our knowledge, we are the first to propose a network model that combines the intermediate hidden layers of BERT and the structural layers of sentences to construct graphs. The contributions of this paper are as follows:(1)A BERT-based hierarchical information encoding method is proposed. The average pooling layer is added to the original BERT layers to extract the semantic information of each subsentence at different layers. Since the encoding of the subsentence is derived from the overall encoding of BERT, the overall semantic information can be considered when encoding the hierarchical information.(2)We propose a novel way of constructing graphs. Our method establishes a hierarchical graph structure based on the hierarchical relationship between BERT layers and sentences and use the graph attention network to extract the hierarchical structure features of the input text to build a multilevel hierarchical relationship graph (directed graph).(3)We propose a novel sentiment analysis model, BHGAttN. The model aggregates semantic features from BERT and sentence structure features after graph training. It not only considers the semantic information, but also pays attention to the structural information between sentences. As a result, BHGAttN can effectively improve the classification performance.(4)We demonstrate that our method outperforms state-of-the-art baseline models through experiments on three datasets.

## 2. Related Work

### 2.1. Text Feature Representation

Sentiment analysis is an important application of text classification. The core problem that determines text classification is text representation, and text vectorization is an important method of text representation. Common methods of text vectorization include discrete representations, such as one-hot encoding, the bag-of-words model (BOW) [19], word frequency-inverse document frequency (TF-IDF) [20], n-gram [21], etc. The characteristics of this method are that the data are high-dimensional and sparse, and the computational complexity of the model is high. At the same time, because the lexical and word order are not considered, the relationship between word vectors cannot be measured, which makes it unable to fully represent different semantic information. The other is based on distributed representation, such as Word2vec [22,23], Glove [24], ELMO [25], GPT [8,9,10], etc. The distributed representation method is based on the language model technology, and the word vector is obtained through the training of the neural network. These methods overcome the limitation of dimensionality and improve the generalization ability of language models, and the obtained text representations can take into account the semantic environment of the context. Recently, based on the large-scale pretrained language model BERT, the model has strong scalability by fine-tuning transfer learning, and has achieved very good results in multiple NLP tasks. In the text, the granularity of extracting information for each layer of BERT is different. In order to introduce more information to enhance the semantic representation, we use the output of hidden states of each layer of BERT to initialize the embedded representation of graph nodes.

### 2.2. Graph Neural Networks

Graph neural networks extract and excavate the features and patterns of graph structure data through the mechanism of message passing. The existing research has proved the effectiveness of the graph-based text classification model. For example, the HR-DGCNN model proposed by Peng et al. [26] regards an article as a graph composed of word nodes and uses a convolutional network of semantic combination to realize topic classification. Zhang et al. [27] proposed to generate a text-level graph model TextING with global parameter sharing for each input text. Compared with the TextGCN [15] model, the TextING model eliminates the dependency burden between corpora, and its performance is better than the graph model built on the whole corpus. Recently, it has been found that the large-scale pretraining model is beneficial to tap the potential of graph learning. Yang et al. [28] proposed GraphFormers, a network architecture that deeply integrates GNN and PLM. This model adopts a hierarchical integration method of GNN and transformer block, enabling interactive training of text representation and graph aggregation. The experimental results show that the prediction accuracy of GraphFormers was greatly improved. Yang et al. [29] combined the advantages of BERT’s semantic encoding and GCN’s structural encoding and proposed a BEGNN model that considered both semantic and structural information and verified the effectiveness of the model on multiple datasets. These methods all build graphs for fine-grained word-level relationships in terms of composition. On the one hand, these models only focus on the short-distance semantic dependencies between words, ignoring the hierarchical relationship between sentences within the sample, which limits the expressiveness of graphs to a certain extent. On the other hand, the combination of GNNs and large-scale pretrained models is limited to shallow feature combinations without deep mining of BERT’s inherent representational capabilities.

Inspired by [30], we consider that the granularity of semantic information extracted by each layer of BERT is different. In the practice of classification tasks, if only the last hidden layer of the BERT model is used as the output, some information, such as phrase-level information, may be lost. Wait. Therefore, our model proposes to utilize information from the intermediate hidden layers of BERT for semantic modeling, while utilizing GAT [31] to map the hierarchical relationships between subsentences in the sample. In this way, the semantic information of sentences can be fully characterized, and the structural relationship between sentences can be captured. Intuitively, our modeling idea can better handle text classification tasks.

## 3. Method

In this section, we describe our modeling approach in detail. First, we show the overall architecture of the model. Second, we detail the specific methods of model implementation.

### 3.1. Model Architecture

The BHGAttN model we designed is shown in Figure 1. The model can be divided into three parts: (1) BERT-based hierarchical information encoding module; (2) GAT-based hierarchical graph network feature extraction; (3) feature fusion classification module. Firstly, the hierarchical relationship between the layers and subsentences of BERT is composed, and a hierarchical relationship graph (directed graph) is established, which considers both the semantics of layers and the hierarchical structure of sentences. The nodes in the graph are subsentence nodes and whole-sentence nodes of each layer of BERT. The features of the subsentence nodes are obtained by encoding the BERT information, and the feature representation of the whole-sentence nodes are obtained by random initialization (the left half of the figure). Nodes between different layers are also connected correspondingly. Secondly, the graph is constructed, it is put into the GAT model for training, and the whole-sentence node representation representing the structural feature is extracted (the right half of the graph). Then, the representations of the whole-sentence node at each level are fused through the attention mechanism to obtain the final hierarchical structure feature representation. Finally, we take out the representation of the first token position output from the last layer of BERT (that is, the overall text semantic representation) and fuse it with the hierarchical structure feature representation extracted through GAT. The implementation method of each module is introduced in following subsections.

### 3.2. BERT-Based Hierarchical Information Coding

Extract the semantic information of each subsentence at different layers. The method is to add a mean pooling layer to the original BERT layers. Figure 2 shows the method for encoding the hierarchical information of BERT.

Suppose a text *S* containing n subsentences is input, denoted as $S=\left\{s_{1},s_{2},\ldots ,s_{n}\right\}$, where $s_{i}$ represents the representation of the *i*-th subsentence. Each subsentence contains at most *l* words, then there is $s_{i}=\left\{w_{i,1},w_{i,2},\ldots ,w_{i,l}\right\}$ for the *i*-th subsentence $s_{i}$. First, take S as a whole, and insert two special characters “[CLS]” and “[SEP]” at the beginning and end, respectively, to indicate the beginning and end of the sentence, so as to process it into an input format suitable for BERT.
(1)$$\hat{S}=\left[\left[\mathrm{CLS}\right],w_{1,1},w_{1,2},\ldots ,w_{1,l},w_{2,1},w_{2,2},\ldots ,w_{2,l},\ldots ,w_{n,1},w_{n,2},\ldots ,w_{n,l},\left[\mathrm{SEP}\right]\right]$$

S^ is then encoded using a BERT with *L* layers. For the *j*-th layer of BERT, the hidden layer representation Hj of S^ in this layer can be obtained:(2)Hj=BERTjS^=hCLSj,h1,1j,h1,2j,…,hn,lj,hSEPj∈ℝnl+2×d
where *d* represents the dimension of the hidden layer vector. In order to obtain the semantic representation of the *i*-th subsentence in the *j*-th layer, the mean pooling operation is applied on the latent vector of the word to which the *i*-th subsentence belongs in the *j*-th layer (as shown by the red box in Figure 1):(3)hij=MeanPoolinghi,1j,hi,2j,…,hi,lj∈ℝ1×d

Furthermore, the hidden layer vector $h_{\left[\mathrm{CLS}\right]}^{L}$ at the $“\left[\mathrm{CLS}\right]”$ position of the last layer of BERT is usually used to represent the overall textual semantics.

### 3.3. GAT-based Hierarchical Graph Network

After obtaining the hierarchical information encoding of each subsentence in Equation (1), this paper constructs a hierarchical graph network $G=\left(V,E\right)$ as shown in the right half of Figure 1, where *V* is the set of nodes and *E* is the set of edges. The network is nested by *L* layers, which correspond to each layer of BERT one by one. *G* contains two types of nodes: subsentence nodes (such as $S_{1}^{1}$) and whole-sentence nodes (such as $S_{ALL}^{1}$). For the same layer, a directed connection is made between the subsentence node and the whole-sentence node, such as $S_{1}^{1}\to S_{ALL}^{1}$; between different layers, a directed connection is made between the node of the previous layer and the corresponding node of the next layer, such as $S_{1}^{1}\to S_{1}^{2},S_{ALL}^{1}\to S_{ALL}^{2}$. Through graph G, the complete hierarchical structure of the sentence is constructed, and the adjacency matrix is shown in Figure 3.

After constructing the above-mentioned hierarchical graph *G*, it is necessary to assign the initial node feature representation *h* to all nodes. For the *i*-th subsentence node in the *j*-th layer, its initial node feature representation can be obtained by Equation (3). For the whole-sentence node, its initial feature representation is obtained by random initialization.

After obtaining the graph and the initialized representation *h* corresponding to the node, the graph attention network GAT is used to extract structural features from the graph. For a GAT with *K* layers and *T* heads, the calculation process can be described as follows:(1)First, GAT needs to calculate the attention weight of each node in the node i and its connected node set in each layer:
(4)$$\alpha_{ij}^{t}=\frac{\exp\left(\mathrm{LeakyReLU}\left(W_{3}^{t}\left[W_{1}^{t}x_{i}^{\left(k\right)}\|W_{2}^{t}x_{j}^{\left(k\right)}\right]\right)\right)}{\sum_{q\in N_{i}}\exp\left(\mathrm{LeakyReLU}\left(W_{3}^{t}\left[W_{1}^{t}x_{i}^{\left(k\right)}\|W_{2}^{t}x_{q}^{\left(k\right)}\right]\right)\right)}$$

In the above formula, *k* represents the *k*-th layer of GAT; *t* represents the *t*-th attention head; $W_{1}^{t},W_{2}^{t},W_{3}^{t}$ are all learnable weight matrices; || represents the concatenation operation.

(2)After the attention weights are obtained, the representation of node *i* in the next layer can be updated by the weighted summation of the neighbor nodes:

(5)$$h_{i}^{\left(k+1\right)}=\begin{array}{c} T\\ \left|\right|\\ t=1 \end{array}\tan h\left(\underset{j\in N_{i}}{\sum }\alpha_{\mathrm{ij}}^{t}W_{4}^{t}h_{j}^{\left(k\right)}\right)$$ where $W_{4}^{t}$ is the learnable weight matrix. After the structural features are extracted by GAT, the representations of all the whole-sentence nodes in the last layer $X_{ALL}=\left[X_{ALL}^{1},X_{ALL}^{2},\ldots ,X_{ALL}^{L}\right]$ are fused through the attention mechanism to obtain the final hierarchical structure feature representation:(6)$$\beta =\mathrm{softmax}\left(W_{\beta }X_{ALL}+b_{\beta }\right)$$
(7)$$H_{\mathrm{Structure}}=\overset{L}{\underset{i=1}{\sum }}\left(\beta_{i}X_{\mathrm{ALL}}^{i}\right)$$
where β is the attention weight, $W_{\beta }$ is the learnable weight matrix, and $\;b_{\beta }\;$ is the bias.

### 3.4. Fusion Classification Module

After obtaining the overall text semantic representation in Equation (1) and the hierarchical structure representation in Equation (2), the two representations are connected and reduced to the classification dimension through a linear layer, and Softmax is used to predict its category:(8)$$p\left(y\right)=softmax\left(W_{y}F+b_{y}\right)$$
where $F=x_{\left[\mathrm{CLS}\right]}^{L}\|X_{\mathrm{Structure}}$. $W_{y}$ is a weight matrix that can be learned. $b_{y}$ is biased. The loss function of model training is cross entropy loss function.
(9)$$loss_{c}=-\overset{N}{\underset{i=1}{\sum }}y^{i}\log p\left(y^{i}\right)$$
where $y^{i}$ represents the true class label of the *i*-th sample.

## 4. Experiment

In this section, we evaluate the effectiveness of the BHGAttN model with extensive experiments on 3 datasets.

### 4.1. Dataset

In our model, since BERT is a multilingual model, it can support multiple languages. In this paper, we test the performance of the model with Chinese, Korean, and English data as representatives. Among them, the English dataset is MR, a movie review dataset commonly used in sentiment analysis. The Chinese and Korean dataset were derived from online comments made by Chinese and Korean netizens on the 2021 Tokyo Olympic Games against the background of COVID-19 via crawlers. For the crawled data, we carried out data cleaning and manual labeling, and randomly sampled 30,000 labeled data according to the ratio of positive and negative labels as the training data of the model. Details of the dataset as shown in Table 1 Statistics of the dataset, including statistics of positive and negative sentiment polarity.

For the above datasets, we randomly split them in a ratio of 8:1:1 for model training, validation, and testing, respectively.

### 4.2. Baseline

To have a clearer comparison, we divide the baseline models into 3 groups for comparative experiments. Details are as follows:

Group 1:Traditional deep learning models based on BERT word embeddings.

In the experiments, we explore initializing the word vectors of the baseline model with BERT to obtain better embedding representation.

BERT-TextCnn [3]: TextCNN is good at short-text feature extraction and is suitable for short-text comment sentiment classification.BERT [12]: An excellent baseline model that performs well on multiple NLP tasks.BERT-BiGRU [32]: In sentiment classification task, bidirectional GRU is used to extract features.BERT-BiGRU_Att [33]: Bidirectional gated recurrent unit (BiGRU) combined with attention mechanism is used to efficiently capture sequence context features.

Group 2:Graph-based text classification models.

This group of experiments mainly compares the effect of different ways of constructing the map on the results.

TextGCN [15]: Huge single-text heterogeneous graph composed of word nodes and document nodes. Transforms a text classification problem into a node classification problem.TextING [27]: Constructs a vocabulary for each document. Whether there is an edge between two words is judged by sliding window method, and all samples are constructed into nodes at a time.BEGNN [29]: Constructs graphs on each text according to word co-occurrence relations, and fuses semantic features acquired by BERT and structural features captured by GCN through co-attention to obtain more effective representations.

Group 3:Ablation experiments.

In this set of experiments, in order to verify the effectiveness of each module of the model, we observe the impact on the results by removing or replacing submodules in the model.

w/o_HGAN: The hierarchical graph network module in the model is removed, the feature representation of each subsentence of the last hidden layer is calculated and connected with the [CLS] token representation after fusion through the attention mechanism, and then classification prediction is performed.w/o_HBERT: The layered coding part based on BERT is removed, only the output of the last hidden layer of BERT is used to encode the graph nodes, and the experimental results are observed.BERT_HGCN: In order to verify the influence of GAT and GCN on the experimental results, we replace the GAT module with GCN for training.RoBERTa_HGAT: We replace the BERT module with Roberta for training and observe the experimental results.

### 4.3. Experimental Setup

In the experiment, the value of the hyperparameter is mainly set according to the previous work experience. In the model architecture, we use BERT-Base as the pretraining language model, which can be well migrated to other transformer based pretraining language models. The dimension of BERT Base hidden layer is 768. The attention of the GAT network is consistent with the number of attention heads in BERT, set to 12, and the dimension of each head is 64. The learning rate is set to 1e-5 and the optimizer is Adam. Dropout is set to 0.5 after the fully connected layer. The epoch of the training is set to 100. For the determination of the maximum number of subsentences, it is set according to the average number of subsentences in different datasets. In addition, in order to better prevent overfitting, we use the early stop method in training. The maximum tolerance for improvement of the f1 value of the validation set in the model is set to 10; that is, the training process is terminated when the performance of the model on the validation set (f1 score) does not improve in 10 consecutive iterations. For the baseline model, we use the same parameter settings as our model, which allows for a fair comparison with our model.

After setting the model parameters, the training, verification, and testing process will be automatically output. If each verification set is improved, a test will be run and the test results will be output. Finally, the performance of the model is subject to the accuracy of the final test set.

### 4.4. Experimental Results and Analysis

In the experimental results (Table 2), we use the accuracy on the test set as an evaluation metric for model performance. From Table 2, we can see that the performance of the graph-based model is overall higher than that of the traditional deep learning model. Our method (the third group) performed the best.

The TextCNN model in the first set of experiments achieves the best performance except for the BERT model with better word embeddings. This is inseparable from its ability to effectively model the semantics of continuous short texts. Similarly, the sequence model BiGRU with pretrained word embeddings also has excellent performance, and the performance of the BiGRU-ATT model of BIRGU with the addition of the attention mechanism has been significantly improved. Compared with traditional deep learning models, the BERT model still maintains the most competitive results. This shows the absolute advantage of large-scale pretrained language models in semantic modeling.

The second group of graph-based methods outperformed the first group overall, indicating that graph networks are effective for text processing. The TextING model improves the composition of the TextGCN model, so that each document has its own graph structure, and the structural information inside the document is well mined. Therefore, its performance is better than the TextGCN model. The BEGNN model, which combines the advantages of the large-scale pretraining model and the TextING model, has achieved the most satisfactory results. This proves that large-scale pretrained models are beneficial to tap the potential of graph learning.

The third group of experiments is our method. Table 2 shows that the BHGAttN model achieves the best results on all datasets. By comparing with the best baseline model, BEGNN, we notice that although the BEGNN model provides the feature interaction module of BERT and GNN, it has a great improvement in performance over other graph models. But our model takes full advantage of the encoded information of the intermediate layers of BERT and introduces a more adequate semantic representation. At the same time, we pay attention to the hierarchical structure between sentences, which can better reflect the structural dependency between document contents than the short-distance word co-occurrence relationship. Experimental results on the MR dataset show that with the same parameter settings, our model outperforms BEGNN by 3.25%, which proves that our method is very effective.

#### 4.4.1. Effectiveness of BERT-Based Hierarchical Information Coding

To examine the impact of different modules in the model on the overall performance of the model, In the experiment, we try to remove the BERT-based hierarchical information encoding module and name it the w/o_HBERT model. We fix the various components of the original model and keep the basic composition of BHGAttN. Only the average pooling layer is added to the output of the last layer of BERT to extract the semantic encoding of each subsentence as the initial representation of the subsentence nodes. The whole-sentence node is obtained by random initialization. After GAT learning, since the number of whole-sentence nodes is 1, we remove the attention feature fusion module, and the model architecture is shown in Figure 4.

We observe from experiments that removing the layered BERT encoding will have a certain negative impact on the results, but our composition method is still better than the method of constructing text graphs based on word co-occurrence relationships. At the same time, it is also illustrated that using the information of the intermediate hidden state of BERT can enhance the semantic features, so that the graph network can obtain better initial embedding representation.

#### 4.4.2. Effectiveness of Hierarchical Graph Attention Networks

We also investigate the impact of hierarchical graph attention networks on model performance. The specific method is to remove the right half of the model architecture (that is, the GAT network module), and name the model after removing the GAT network module w/o_HGAT (Figure 5). Similarly, we use the output of the last hidden state of BERT and obtain the representation of each subsentence through mean pooling, and then use the attention mechanism to fuse the representation of each subsentence. Finally, it is concatenated with the output of the [CLS] token of the last layer of BERT and mapped to the classification dimension through a linear layer for classification.

During the experiment, we noticed that the performance of the model was degraded by removing the GAT layered network module, indicating the effectiveness of the layered graph network in the model. However, at the same time, the experimental results show that w/o_HGAT is still higher than the [CLS] classification performance of the BERT model. It is proved that the subsentence feature representation can promote the performance of the model.

#### 4.4.3. Influence of Different Graph Networks on Experimental Results

We also compare the effect of layered GAT composition and layered GCN composition on model performance. Replace the GAT network on the right part of the model architecture with GCN. The experimental results in Table 2 show that GCN-based models perform slightly lower than GAT. The possible reason is that when the GCN model learns the node representation, the weights of the edges are fixed, which limits the expressive ability of the edges to a certain extent. GAT, on the other hand, adaptively learns the edge weights through the attention mechanism, which makes it more effective at fusing the information of node features and graph structure.

#### 4.4.4. Influence of Large-Scale Pretraining Model on Experimental Results

In order to further verify the advantages brought by the large-scale pretrained language model, we replaced the BERT pretrained language model with RoBERTa [34] for experiments. From the experimental results in Table 2, it can be seen that the performance of RoBERTa-HGAN is much improved than that of BHGAttN, which is because RoBERTa improves the pretraining method of BERT, which makes RoBERTa outperform BERT. It can be seen that an excellent large-scale pretraining model is beneficial to the model.

## 5. Conclusions

In this paper, in view of the problems existing in the current text classification task, we propose to make full use of the advantages of large-scale pretraining models and graph neural networks and design a text classification model based on BERT and GAT to model hierarchical relationships. Different from previous work, on the one hand, we propose a BERT-based hierarchical information encoding method to enhance semantic features through the output of the intermediate hidden state of BERT; On the other hand, a hierarchical graph network corresponding to each layer of BERT is constructed, and the graph attention network is used to extract the hierarchical structure features of the input text. The model we constructed considers both the semantic features of layer-by-layer text and the hierarchical relationship between text contents. The experimental results demonstrate the effectiveness of the model. However, due to the huge number of parameters in the large-scale pretraining model, the method of constructing the graph layer-by-layer will increase the number of edges in the graph, which increases the burden of memory.

Therefore, in future work, we will explore how to further improve the performance of the model with low memory consumption. For example, we consider using a lightweight pretraining language model to replace BERT. Greatly reducing the number of parameters can reduce the calculation cost and greatly improve the training speed of the model. In addition, the cause of memory overload is closely related to the graph size. In the next step, we will dig deeper into the linguistic information encoded in the neural network model. Specifically, it is to explore the different performance of hidden layers in the middle and analyze which layers have more positive impact on semantic encoding. When modeling, we will consider removing less influential layers and reducing the size of the graph to some extent to reduce memory consumption.

## Figures and Tables

**Figure 1 entropy-24-01691-f001:**
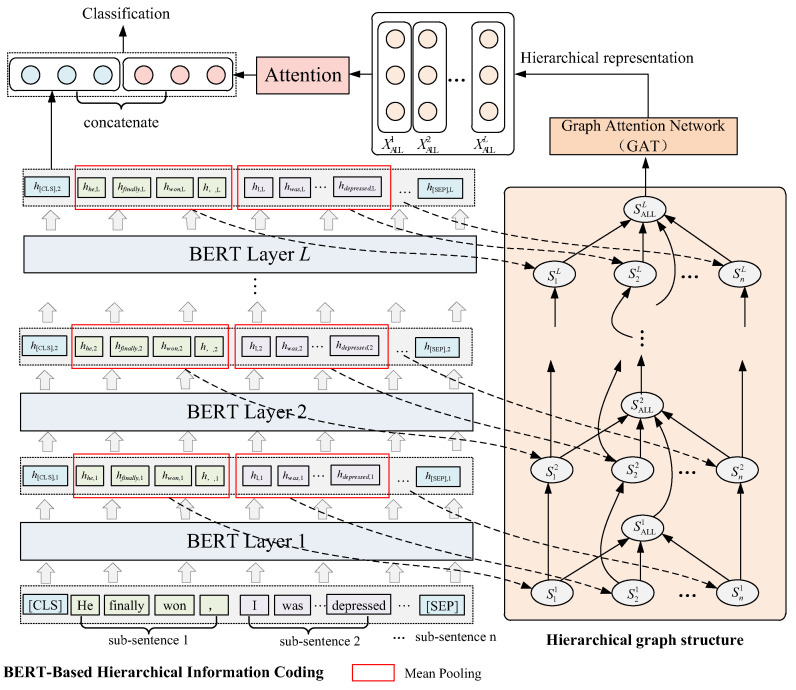
Architecture of BHGAN model. The left half of the figure is the BERT-based hierarchical encoding, and the right half is the graph construction and GAT-based feature extraction. The above is to use the representation of the overall nodes at each level through the attention mechanism to obtain the final hierarchical structure representation. The red border represents the mean pooling operation on the latent vector.

**Figure 2 entropy-24-01691-f002:**
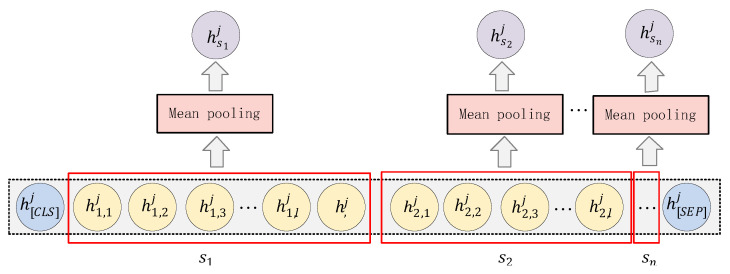
BERT-based information encoding.

**Figure 3 entropy-24-01691-f003:**
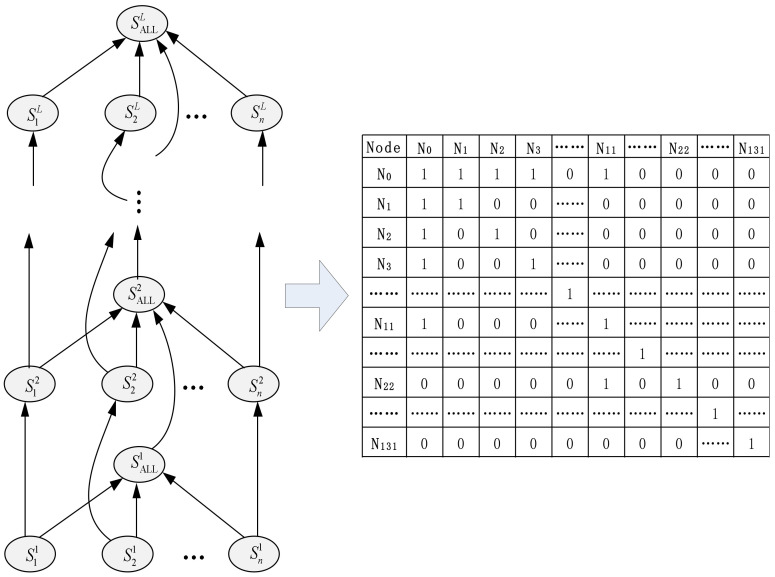
Adjacency matrix. For example, for a sample containing 7 subsentences, first expand the number of subsentences to the maximum number of subsentences (set to 10 in this paper) and pad in 0 if the number is not enough. If 12 layers of BERT are used, including whole-sentence nodes, there are 10+1∗12=132 nodes in total. Each node is represented by Nii =0,1,2,⋯131. Adjacency matrix A∈ℝV×V, where V is the number of nodes.

**Figure 4 entropy-24-01691-f004:**
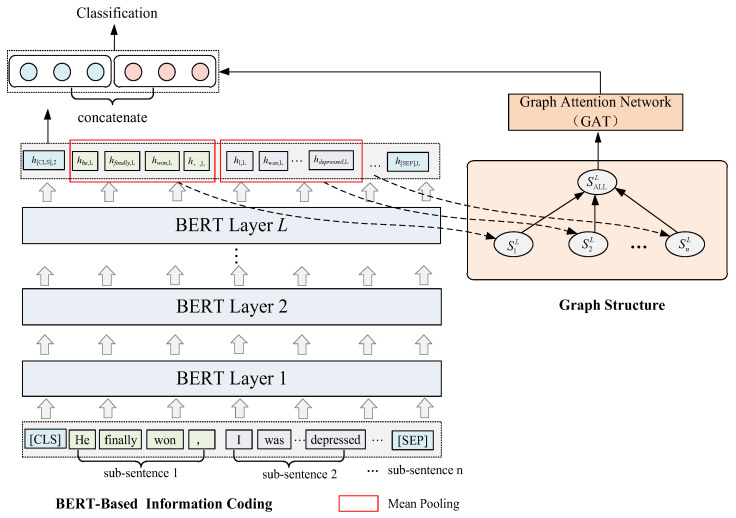
w/o_HBERT model architecture.

**Figure 5 entropy-24-01691-f005:**
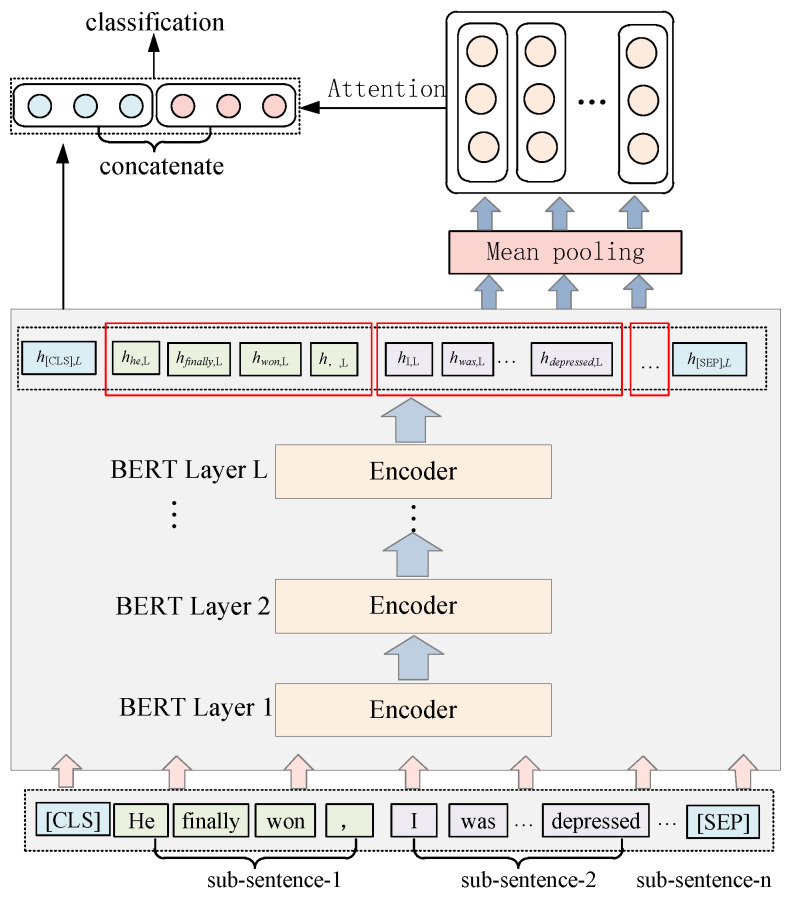
w/o_HGAT model architecture.

**Table 1 entropy-24-01691-t001:** Statistics of the dataset, including statistics of positive and negative sentiment polarity.

Dataset	Positive	Negative	Total
Ch_TOR (Chinese Olympic Review)	15,000	15,000	30,000
Ko_TOR (Korean Olympic Review)	5571	24,429	30,000
MR (Movie Review)	3554	7108	10,662

**Table 2 entropy-24-01691-t002:** Test accuracy (%) of each model on 3 datasets.

**Category**	**Model**	**Dataset**		
		**Ch_TOR**	**Ko_TOR**	**MR**
BERT-based traditional model	BERT_TextCNN	77.60	86.56	77.69
	BERT_BiGRU	76.13	84.86	76.1
	BERT_BiGRU-Att	76.67	86.12	77.32
	BERT	81.83	86.62	86.22
Graph-based model	TextGCN	-	-	76.74
	TextING	-	-	78.74
	BEGNN	-	-	84.47
Ablation experiment (Ours)	**BHGAttN**	**82.63**	**87.79**	**87.72**
	w/o_HBERT	81.96	86.63	86.68
	w/o_HGAT	82.26	87.06	86.22
	BERT_HGCN	82.03	87.29	86.32
	**RoBERTa_HGAT**	**83.27**	**-**	**88.85**

## Data Availability

The data provided in this study is collected with crawler programs from the online comments in TOG news from Naver News Network (www.news.naver.com) in South Korea, Sina News Network (https://news.sina.com.cn) in China, and New York Times (https://www.nytimes.com/) in the United States. The download URL of the public dataset MR is https://github.com/CRIPAC-DIG/TextING (accessed on 15 November 2022).

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
