# Peer review of "BHGAttN: A Feature-Enhanced Hierarchical Graph Attention Network for Sentiment Analysis"

_entropy, 2022, doi:10.3390/e24111691_

Round 1

Reviewer 1 Report

This is quite an interesting paper on the approach for the graph attention network to model the hierarchical relationship of texts. The only concern about the results: authors should provide an extra explanation for choosing the dataset. Will this method work on other data?  How can it be generalized to other languages?

Reviewer 2 Report

The manuscript proposes a novel modeling approach based on the BERT model and a hierarchical graph network. The authors adopted the hierarchical semantic information of BERT and built a multi-layer relationship graph that focuses on layer-by-layer, extracting semantic information from the outputs of the hidden state of the BERT model.

The paper is well-written and structured, and the idea sounds potentially interesting in text classification tasks. My only concerns regard the first part of the paper, in which the authors should better emphasize the novelty and clearly indicate the points of innovation in the presented solution. Why is this unique warrant for readers' attention? How does the manuscript contribute to the body of knowledge? In my opinion, the discussions should be elaborated and well-positioned with respect to similar published literature. Furthermore, the authors should state the justification of the test methods, parameters, and sampling dataset.

Moreover, the experiments regard only a sentiment analysis task. The title of the manuscript focuses on Text Classification, but it seems too assertive if tested on a more vertical task. At least a discussion on future experiments and a preliminary insight or expectations should be inserted in the manuscript. Otherwise, I suggest focusing on sentiment analysis in the title to strengthen the manuscript scientifically and be more motivated by the experiments.

Finally, as in the Conclusions section the author state that constructing the graph layer by layer increase the burden of memory, it could be helpful to have a quantitive comparison between the average computational time requested by the proposed system and the average computational time of the baseline systems. Moreover, the authors should summarize how they will improve the model to have low memory consumption.

Minor points and typos:

Line 16

- a hierarchical graph network model BHGAttN based on a large-scale pre-trained model --> BHGAttN should be placed within brackets.

Line 104-105

There is a missing reference

Line 118

- The existing research have proved --> The existing research has proved

Line 170

- features containing textual contextual semantic information --> this sentence is not clear...

Line 208

- Missing space before ""Through""

Line 401-402

- However, due to the huge amount of parameters of the large-scale pre-training model. --> The sentence is concluded but there is an open concept. Maybe the authors intended to put a comma instead of a point. 

Round 2

Reviewer 2 Report

The revised manuscript fulfilled all the requested improvements, and the issues recognized in the previous submission have been adequately addressed and solved. I would like to thank the authors for this revised work, and I suggest to accept the paper in the current version.